# Development of a BiAD Sensor for Locus-Specific Detection of Cellular Histone Acetylation Dynamics by Fluorescence Microscopy

**DOI:** 10.3390/genes16040444

**Published:** 2025-04-10

**Authors:** Anja R. Köhler, Nicole Gutekunst, Annika Harsch, Pavel Bashtrykov, Albert Jeltsch

**Affiliations:** Institute of Biochemistry, University of Stuttgart, Allmandring 31, 70569 Stuttgart, Germany

**Keywords:** epigenome modification, histone acetylation, fluorescence microscopy, split fluorophore, bimolecular fluorescence complementation, bimolecular anchor/detector, single-cell analysis

## Abstract

Background: Dynamic changes in histone acetylation play crucial roles during cellular differentiation and disease development, but their detection in living cells is still a challenging task. Objectives: Here, we developed a Bimolecular Anchor Detector (BiAD) sensor for the detection of locus-specific changes in histone acetylation in living cells by fluorescence microscopy. Methods: We used the BRD9 bromodomain cloned as tandem double domain (2xBRD9-BD) as a reader of histone acetylation. It was integrated into a dual-color BiAD chassis that was previously described by us. Results: We identified the gene body of *TTC34* as a potential target for our sensor, because it contains dense histone acetylation and 392 local sequence repeats. Using a binding-deficient mutant of 2xBRD9-BD as a negative control, we established a successful readout of histone acetylation at the *TTC34* locus. A single-domain reader did not function, indicating the requirement for the double reader to enhance the affinity and specificity of the chromatin interaction via avidity effects. With this sensor, we could detect dynamic increases in histone acetylation at the *TTC34* locus after the treatment of cells with the histone deacetylase inhibitor Trichostatin A for 6 h indicating the applicability of the sensor for single-cell epigenome studies. Conclusions: Our data demonstrate that active chromatin modifications can be detected by BiAD sensors using 2xBRD9-BD as a reader. This complements the toolkit of the available BiAD sensors and documents the modularity of BiAD sensors.

## 1. Introduction

Histone lysine acetylation is an integral part of the epigenome and central to the epigenetic control of gene transcription [1,2]. Chromatin modifications are bound by special reader proteins which trigger downstream signaling events [3]. Various reader proteins have been used in biotechnological applications to develop fluorescence microscopy sensors for the detection of epigenome modifications in cells [4]. The bromodomain (BD) has been found in many chromatin-associated proteins and enzymes and shown to function as a general reader domain for acetyllysine residues on histones and other proteins [5,6]. BD-containing proteins have been fused to fluorophores and used for the global detection of histone acetylation in cells for many years, and the corresponding sensor systems have been continuously improved [7,8,9,10]. However, all the established cellular sensor systems so far do not permit detection of histone acetylation at defined genomic loci. Hence, global histone acetylation changes can be observed with these sensors in living cells, but the findings cannot be connected to individual genes which limits the applicability of these sensors. On the contrary, analysis of histone acetylation by ChIP-seq (and variants thereof) [11] requires the lysis of cells prevailing the direct association of specific acetylation patterns with cellular phenotypes. Hence, novel techniques for the detection of histone acetylation with locus resolution in living cells are needed.

During the last years, we developed Bimolecular Anchor Detector (BiAD) sensors to overcome this limitation and allow for a locus-specific readout of epigenome modifications in living cells [12,13]. In principle, the BiAD sensors comprise a programmable DNA-binding anchor module for locus-specific targeting of the sensor, which is a single-guide RNA (sgRNA)/dCas9 (deactivated Cas9) complex in the most recent applications. This module is combined with chromatin reader domains used as detector modules for the recognition of defined chromatin modifications at the genomic target locus. Both modules are fused to the complementary parts of a split fluorophore, which leads to bimolecular fluorescence complementation when both modules bind in close spatial proximity indicating that the target locus contains the chromatin modification of interest [12]. In a recent work, the first generation BiAD sensors (described in [12]) were massively improved [13] in three aspects: (1) To enhance the sensitivity of BiAD sensors, a signal amplification using the 10xSunTag was implemented which enabled the recruitment of multiple detector modules to a single sgRNA/dCas9 binding site. (2) The epigenetic readout was combined with a second YPet fluorophore which is recruited by the sgRNA and labels the locus of interest. The resulting dual-color BiAD sensors allow to detect the BiAD signal specifically at the locus of interest, which led to a strong improvement of signal-to-noise ratio, because background fluorescence fluctuation and non-specific protein aggregation could be excluded from the analysis. (3) As a third improvement, it was demonstrated that double-reader domains generally perform better in BiAD applications, finally leading to a system that allows the detection of DNA methylation, as well as di- and tri-methylation of H3K9, H3K27 and H3K36 at endogenous loci with a minimum of 45 local repeats.

BiAD sensors have been used to detect the dynamics of DNA and H3K9 methylation upon DNA or protein methyltransferase KO or KD, or after treatment of cells with a methyltransferase inhibitor [12,13]. Moreover, elevated H3K9 methylation signals could be detected at the inactive X-chromosome [13], locus-specific reductions of H3K9me3 at mouse major satellite repeats were found in p53 deficient cells [14], and changes in human α-satellite repeat DNA methylation were detected depending on the density of culturing [15]. All these biological applications demonstrate the wide applicability of the BiAD sensors, even at their current state of development being able to detect epigenome modifications only at regions with a high-to-moderate level of local sequence repetitions. However, so far, no BiAD sensor has been made available for active chromatin modifications, like H3K4me3 or histone acetylation. It was the aim of this work to fill this gap by developing and validating a functional dual-color BiAD sensor for the detection of histone acetylation at defined genomic loci in living cells. For this, an acetyllysine reader domain needed to be developed, integrated into the existing dual-color BiAD chassis, and the functionality of the BiAD sensor for histone acetylation needed to be validated. Finally, we aimed for a demonstration of the application of the novel BiAD sensor to detect dynamic changes of histone acetylation at a defined target locus in living cells.

## 2. Materials and Methods

### 2.1. Cloning of the Acetyllysine Detector Domains

The bromodomain of BRD9 (Uniprot Q9H8M2-1, amino acids 14–134) was amplified from HEK293 cDNA and cloned into the BiAD detector-IFP2.0C vector plasmid [13] using Gibson Assembly. To create a binding-deficient detector mutant, the conserved tyrosine residue Y57 was substituted with alanine via site-directed mutagenesis, because the Y-to-A substitution of the corresponding residue in BRD2 (Y113, Uniprot P25440-1) was shown to disrupt the H4K12ac binding [7,9,16]. The double-domain construct, in which both domains are separated by a 12-amino-acid linker containing an SV40 large T-antigen monopartite NLS was cloned as described [13]. Otherwise, the constructs described in our previous study were used including the vector expressing the GATGCTCACCTG sgRNA [13].

### 2.2. Cell Culture

Human embryonic kidney cells (HEK293, RRID: CVCL_0045) were sourced from DSMZ (Braunschweig, Germany). The cells were cultured in DMEM (Dulbecco’s Modified Eagle Medium) supplemented with 10% fetal bovine serum (FBS), 1% penicillin/streptomycin and 4 mM L-glutamine, and maintained at 37 °C with 5% CO_2_ in a humidified incubator (BINDER). To sustain a confluency of 70–90%, the cells were sub-cultured at a 1:7 to 1:10 ratio every 2 to 3 days. For this, the cells were washed with PBS (lacking CaCl_2_ and MgCl_2_), followed by the addition of trypsin-EDTA solution (Merck KGaA, Darmstadt, Germany) and incubation at 37 °C until the cells detached. Afterwards, the cells were resuspended in a fresh culture medium and split at the required ratio. For cryopreservation, cells were pelleted at 300 g for 5 min, resuspended in a freezing medium (90% FBS and 10% DMSO) and gradually frozen to −80 °C.

### 2.3. Transfections

For transient transfections, HEK293 cells were seeded on glass cover slips in a 6-well plate at a density of 200,000 cells per well one day before transfection. Right before transfection, the medium was exchanged with a 1.5 mL fresh culture medium supplemented with 30 nM biliverdin which serves as the chromophore for IFP2.0. Cells were transfected with 400 ng sgRNA, 100 ng dCas9-10xSunTag, 300 ng scFv-IFP2.0N, 300 ng BRD9BD-IFP2.0C/2xBRD9BD-IFP2.0C and 400 ng MCP-YPet using polyethylenimine (PEI) at a ratio of 1:3 as the transfection reagent. Cells were fixed with 4% paraformaldehyde 24 h after transfection as described in [13].

### 2.4. HDAC Inhibitor Treatment

HEK239 cells were transfected as described in the previous section and treated with 5 µM Trichostatin A (TSA, PubChem compound **444732**) (Merck KGaA, Darmstadt, Germany) or an equivalent amount of DMSO for six or three hours before fixation.

For validation of the TSA treatment, histones were purified by acid extraction [17]. In short, ~2.5 × 10^6^ cells were pelleted at 300× *g* for 5 min and washed twice with PBS. Cells were resuspended in a lysis buffer (1 mM KCl, 1.5 mM MgCl_2_, 1 mM DTT, 10 mM Tris-HCl pH 8.0) and incubated for 30 min at 4 °C on a rotator. Cells were centrifuged for 10 min at 10,000× *g*, and the supernatant was discarded. The pellets were resuspended in 0.4 M H_2_SO_4_ and incubated on a rotator overnight. On the next day, the samples were centrifuged for 10 min at 16,000× *g*, and the supernatant containing the histones was transferred into fresh low-binding tubes. TCA precipitation was performed for 30 min on ice. Samples were centrifuged for 10 min at 16,000× *g*. The protein pellets were washed twice with ice-cold acetone with 5 min 16,000× *g* centrifugation steps between each washing. All these steps were conducted at 4 °C. The pellets were air-dried for 20 min at RT and resuspended in 80 µL H_2_O.

Purified histone protein samples were separated on a 15% SDS-PAGE. The proteins were visualized either by Coomassie BB staining, or they were transferred to a nitrocellulose membrane, blocked with 5% milk in PBST and incubated with H4 pan-acetyl antibody (Active motif Kat. No. 39243, RRID: **AB_2793201**) in a dilution of 1:750 overnight at 4 °C. This was followed by washing with PBST, 1 h of incubation with the anti-rabbit secondary antibody coupled to horseradish peroxidase (Cytiva NA934) in a 1:5000 dilution at room temperature and washing with PBST. The Western blot was developed using SuperSignal™ West Femto Maximum Sensitivity Substrate (Thermo Scientific, FEI Deutschland GmbH, Dreieich, Germany), and the image was captured with Fusion Advance detection system (VWR International GmbH, Darmstadt, Germany).

### 2.5. Fluorescence Microscopy and Image Analysis

The fixed cell samples were imaged using a confocal laser scanning microscope (LSM710 or LSM980 Airyscan 2, Carl Zeiss Microscopy GmbH, Jena, Germany) using the image acquisition settings described [13]. The settings were kept constant within the same experiment. The stacks were superimposed to generate maximum intensity projections using the ZEN 3.0 SR (black edition) software (Carl Zeiss Microscopy GmbH, Jena, Germany). Image export was performed using the ZEN 3.0 (blue edition) software. The exemplary images in each figure were exported with identical brightness and contrast settings.

Images were analyzed using a custom FIJI (ImageJ 1.54b) [18] macro without adjustments of contrast and brightness as described [13]. In brief, two intensity thresholds were set manually for each cell to define the nucleus and spots as regions of interest (ROIs) based on the signal in the YPet channel. The mean intensity of the ROIs was measured across all channels. For quantitative analysis, the average nuclear background signal intensity was subtracted from the spot intensities in each channel. The BiAD spot intensity (IFP2.0 channel) was then normalized to the corresponding YPet spot intensity, and the normalized BiAD signals from all spots within a single cell were averaged and are represented as a dot in the boxplots. Boxplots were created using the Seaborn Python data visualization library [19,20].

## 3. Results

BiAD sensors fill an important gap in the available technologies by allowing the detection of epigenome modifications with locus specificity in single living cells by fluorescence microscopy [12,13]. The most recently developed dual-color BiAD sensors [13] combine two fluorescence signals and two orthogonal signal amplification systems (Figure 1A). The target locus is bound by a sgRNA/dCas9 complex, and its location is indicated by the recruitment of several YPet fluorophores to the sgRNA via the MS2 system [21]. The readout of the epigenome modification is based on the reconstitution of the N- and C-terminal parts of the split IFP2.0 fluorophore [22,23]. The N-terminal part of IFP2.0 is fused to a single-chain variable fragment (scFv) antibody domain binding to the GCN4 peptide epitope which is present in 10 copies in the so called 10xSunTag fused to the dCas9 [24]. The C-terminal part of IFP2.0 is fused to an appropriate epigenome reader domain for detection of the epigenome modification of interest at the target locus. In the past years, BiAD sensors have been developed for the locus-specific detection of DNA methylation as well as H3K9, K27 and K36 methylation, but so far, a detector for an active chromatin mark is missing in the BiAD toolbox. Here, we aimed to develop and validate such a novel tool for the detection of histone acetylation.

### 3.1. Identification of a Target Region for Histone Acetylation Detection with the Dual-Color BiAD Sensor

Despite their enhanced sensitivity, even the most developed second generation dual-color BiAD sensors require the presence of at least 45 local DNA repeats at the target region allowing for the binding of several sgRNA/dCas9 complexes to generate visible BiAD signals [13]. This requirement entailed a critical problem, because repetitive sequences in the human genome usually are epigenetically silenced. Therefore, there were no obvious candidate regions containing histone acetylation at local repeats that could be used as a target for the development of a BiAD sensor for histone acetylation. To search for a suitable target region, we resorted to a list of repetitive regions containing potential sgRNA binding sites in the human genome [21] which was compared with available ChIP-seq data for HEK293 cells [25]. HEK293 cells were selected for the development of the BiAD sensors, because of their easy handling and high transfectability. This comparison identified the gene body of *TTC34* as a potential target for the measurement of histone acetylation with a BiAD sensor, as it contains 392 local repeats of a CCACAGGTGAGCATC sequence [21] and dense histone acetylation (Figure 1B).

### 3.2. Integration of a Functional Acetyllysine Reader Domain into the Dual-Color BiAD Chassis

As described above, bromodomains have been identified as readers of acetyllysine [5,6]. More specifically, the bromodomain of BRD9 was identified in peptide spot-binding experiments as pan-histone acetylation detector [26]. Based on different, previous studies from other labs [10,27,28,29] and our own data [13,30], we expected that the binding affinity and specificity of the BRD9-BD as a detector module toward histone acetylation would be improved by using two BDs fused in a tandem arrangement into a double domain, called 2xBRD9-BD. The BRD9-BD was cloned from human cDNA, and the single and double domain integrated into the existing dual-color BiAD sensor addressing the *TTC34* repeats [13]. To generate a binding-deficient BRD9-BD as a negative control, we mutated the conserved tyrosine residue Y57 in the BRD9-BD and 2xBRD9-BD context to alanine, because the corresponding Y113A mutation in bromodomain 1 (BD1) of the BRD2 protein (Appendix A) was shown to disrupt H4K12ac binding [7,9,16].

HEK293 cells were transfected with all components of the dual-color BiAD sensor for histone acetylation detection at the *TTC34* target locus with either the wildtype 2xBRD9BD detector module (WT) or the Y57A mutant. The cells were inspected by fluorescence microscopy 24 h after transfection. They showed strong colocalization of the BiAD signal (IFP2.0) with the locus marker fluorophore (YPet) for the WT detector, but not for the binding-deficient Y57A mutant (Figure 2A). A quantitative analysis revealed a highly significant increase in the BiAD signal at the target loci in all three independent experiments with intact (WT) detector modules but not with the mutant detector modules (Figure 2B,C). These data validate the function of the novel BiAD sensor for the detection of histone acetylation. Similar experiments with the BRD9 single BD did not result in a visible or statistically significant BiAD signal increase at target loci with the WT reader domain when compared with the inactive mutant, indicating that the single-BD BiAD reader is not functional (Appendix A). Hence, double reader domains improve the functionality of the BiAD sensor though avidity effects as previously observed for the HP1ß-chromodomain, DNMT3A-PWWP domain and CBX7-chromodomain readers of H3K9, H3K36 and H3K27 di- and tri-methylation [13].

### 3.3. Application of the Novel BiAD Reader to Detect Dynamic Changes of Histone Acetylation in Living Cells

Next, we aimed to apply the newly developed BiAD sensor for histone acetylation at the *TTC34* locus to detect dynamic changes of this modification in living cells. To this end, HEK293 cells were transfected with the plasmids encoding the sensor components and either treated with TSA for 3 and 6 h or treated with DMSO as a negative control. TSA is a well-established inhibitor of class I and II histone deacetylases [31]. When applied to the cell culture medium, it leads to a strong increase in histone acetylation. For validation of the TSA treatment, histone proteins were extracted. Histone acetylation was analyzed by Western blot using α-H4 pan-acetyl Ab as expected showing an about 10-fold increase in the histone acetylation signal after the TSA treatment, (Appendix A). As in the previous analysis with untreated cells (Figure 2), the WT sensor detected a stronger BiAD signal than the Y57A mutant sensor (Figure 3A) with a *p*-value of 6.0 × 10^−3^ in the DMSO-treated control cells (Figure 3D). However, comparison of the BiAD signals of the DMSO-treated sample and cells treated with TSA for 3 h (Figure 3B) and particularly for 6 h (Figure 3C) indicated a clear and statistically highly significant increase in the BiAD signal in the case of the intact sensor system but not with the system including the binding-deficient Y57A BD (Figure 3D and Appendix A).

## 4. Discussion

In this work, we developed a dual-color BiAD sensor for the detection of histone acetylation, described its validation and demonstrated an exemplary application of the novel sensor for the detection of histone acetylation at the *TTC34* locus. The functionality of the newly developed BiAD sensor for histone acetylation is documented by these experimental observations:Histone acetylation at the *TTC34* locus has been experimentally shown in the cell line used in our experiments.Acetyl-specific binding of BRD9 and its loss by the Y57A mutation is well documented in many papers.We observed specific BiAD signals with the WT sensor at the *TTC34* locus but not with the completely identical negative control BiAD sensor that just differs by a single Y57A mutation in both domains of the 2xBRD9 double reader.We observed an increase in global histone acetylation levels after the TSA treatment.Our BiAD sensor (but not the mutant control sensor) detected an increase in histone acetylation at the *TTC34* locus after the TSA treatment.

Our data demonstrate that active chromatin modifications can be detected by BiAD sensors which complements the toolkit of available BiAD sensors for repressive chromatin modifications. The results of our study confirm the modularity of BiAD sensors [12,13] suggesting that additional readers for even more epigenome modifications could be added in a similar fashion if an appropriate reader domain and target locus can be identified. Based on our experience with previous dual-color BiAD sensors [13], the newly developed sensor for histone acetylation is fully compatible with live cell imaging, and it will also allow to capture videos over several hours.

However, while demonstrating the clear capability of locus-specific detection of histone acetylation at the *TTC34* gene body, the BiAD sensor developed here suffers from a similar problem as its predecessors described in Köhler et al. 2024 [13]. This is illustrated by the observation in Figure 2 that even in the system using the WT 2xBRD9-BD detector module, many cells apparently were BiAD negative. While one cannot rule out cellular heterogeneity as a reason for this variability, there is an important technical issue to be considered, which is that the BiAD sensor used here depends on the co-transfection of five individual plasmids. These plasmids encode dCas9-10xSunTag (plasmid 1), MCP-YPet (plasmid 2), sgRNAs (plasmid 3), scFv fused to the N-terminal part of IFP2.0 (plasmid 4) and the fusion of reader domain with the C-terminal part of IFP2.0 (plasmid 5). Among them, successful transfection of plasmids 1–3 can be validated by the appearance of the YPet signal at the target locus, allowing to restrict the analysis on cells that definitely contain these 3 plasmids. However, it remains invisible if target cells really contain plasmids 4 and 5, and given the limitations of co-transfections, it is likely that some of the apparently BiAD negative cells observed with the WT 2xBRD9-BD BiAD sensor in reality were “false negatives”, because they did not contain one of these plasmids, and hence did not harbor a functional BiAD sensor. This problem could be overcome in future work by the generation of stable cell lines expressing the BiAD components encoded by plasmids 4 and 5 or by strategically combining multiple BiAD components on the same plasmid to make the successful transfection of all required plasmids visible.

Such a conceptional improvement should considerably increase the dynamic range between the signals observed with WT and binding-deficient negative control BiAD sensors. This would directly enhance the sensitivity of the sensor. Whether such improvement of the sensitivity would be sufficient to reach single-locus resolution which would allow us to use the sensor at any gene, remains to be tested experimentally. Moreover, the elimination of false negative cells from the analysis would facilitate true single-cell analysis of the BiAD signals by removing the need to analyze entire cell populations for statistical purposes in order to draw conclusions about individual cells. In such an experimental setting, single living cells could be tracked over time to directly correlate changes in the locus-specific epigenomic landscape to alterations in the cellular phenotype. It will be interesting to apply this sensor to additional target loci in future work. Another interesting future aim could be to combine two or more BiAD sensors in one cell, allowing the dynamic detection of more than one epigenome modification at the same time.

## Figures and Tables

**Figure 1 genes-16-00444-f001:**
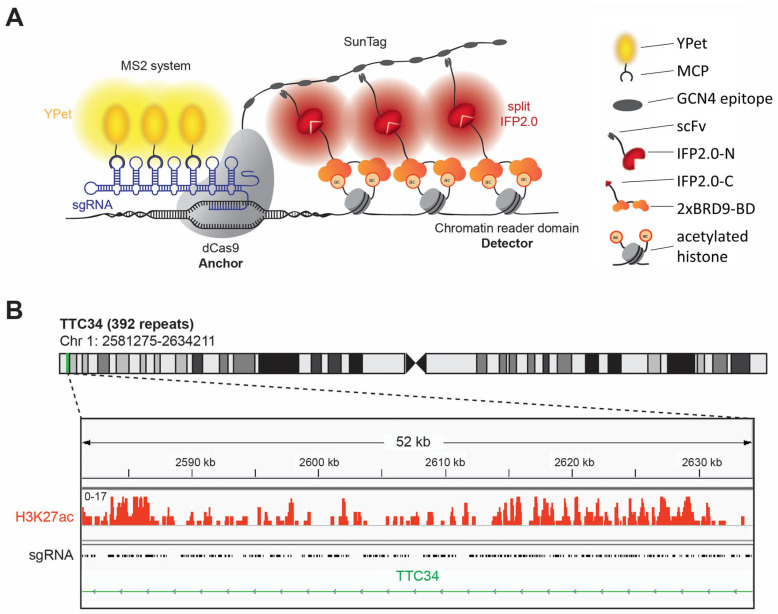
Development of a dual-color BiAD sensor for detection of histone acetylation. (**A**) Schematic representation of the sensor design. The target locus is visualized by a full-length fluorophore (YPet), recruited by an MS2 scaffold of the sgRNA in complex with the anchor module (dCas9). One part of the split fluorophore (IFP2.0) is recruited to the SunTag amplification system fused to dCas9 by a single-chain variable fragment (scFv) antibody domain. To detect acetylated histones, a double BRD9 bromodomain (2xBRD9-BD) is fused to the second part of the split IFP2.0 fluorophore. If histone acetylation (Kac) is present at the target locus, the detector module binds at this place bringing the second part of the split IFP2.0 in close spatial proximity to the first one leading to reconstitution of the split IFP2.0 and appearance of fluorescent BiAD signal. (**B**) Identification of a repetitive genomic region containing histone acetylation, which can be used for the validation of the novel detector module. The gene body of the *TTC34* gene contains 392 repeats of the sgRNA binding sites (indicated in the black trace), and it is modified with H3K27ac (indicated in the orange trace).

**Figure 2 genes-16-00444-f002:**
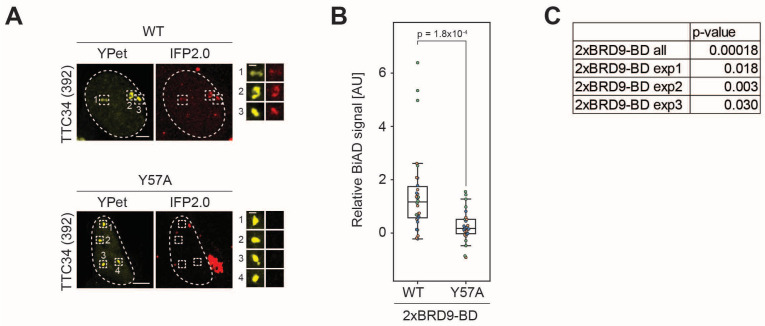
Validation of the BRD9-BD double domain (2xBRD9-BD) for histone acetylation readout at the *TTC34* locus. HEK293 cells were transfected with all components of the dual-color BiAD sensor for histone acetylation detection at the *TTC34* target locus with either the wildtype 2xBRD9-BD detector (WT) or a binding-deficient mutant (Y57A). (**A**) Exemplary fluorescence microscopy images showing strong BiAD signal (IFP2.0) at the marker fluorophore (YPet) spots with the WT detector, but not for the negative control sensor containing the binding-deficient Y57A mutant. Scale bars are 5 μm and 1 μm for the magnified images. Cell nuclei are indicated by dotted lines. Total number of analyzed cells: WT 36, mutant 35. (**B**) Boxplot showing the relative BiAD signals of three independent experiments (depicted in orange, blue and green) normalized as described in the Methods section. Significance was determined via a two-tailed, unpaired *t*-test. (**C**) *p*-values between WT and Y57A for all three combined data sets as well as for each of the individual experiments. See also Appendix A.

**Figure 3 genes-16-00444-f003:**
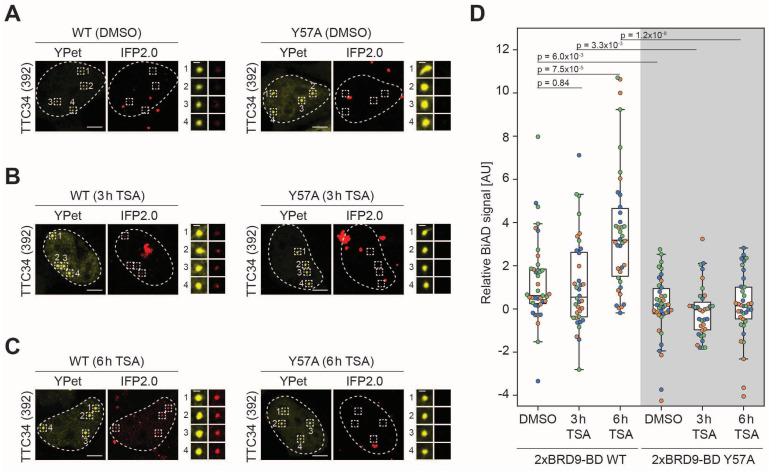
Visualization of locus-specific gain in histone acetylation upon TSA treatment. HEK293 cells were transfected with all components of the dual-color BiAD sensor for histone acetylation detection at the *TTC34* target locus with either the wildtype 2xBRD9-BD detector (WT) or a binding-deficient mutant (Y57A). Cells were either mock-treated (DMSO) or treated with TSA for 3 h or 6 h before fixation. (**A**–**C**) Exemplary fluorescence microscopy images showing appearance of the BiAD signal (IFP2.0) at the marker fluorophore (YPet) regions for the WT detector but not for the binding-deficient Y57A mutant. The cells were either treated with DMSO (**A**) for 3, (**B**) or 6 h with TSA (**C**). Cell nuclei are indicated by dotted lines. Total number of analyzed cells: WT A/B/C 39/36/38, mutant A/B/C 38/35/37. (**D**) Boxplot showing the relative BiAD signals of three independent experiments (depicted in orange, blue and green) normalized as described in the Methods section. The significance of differences of the BiAD signals was determined via a two-tailed, unpaired *t*-test. *p*-values are indicated in the boxplot. The WT BiAD sensor showed an increase in the IFP2.0 signal in cells after 6 h treatment with TSA compared to the DMSO negative control. Additional pairwise *p*-values are provided in Appendix A.

## Data Availability

All analyzed data are included in the published article and its Appendix A. The plasmids encoding the 2xBRD9-BD fused to the C-terminal part of IFP2.0 were deposited with Addgene (85493). The other previously described parts of the dual color BiAD system are available at Addgene (83849). The ImageJ macro used for image analysis is available at Figshare https://doi.org/10.6084/m9.figshare.23592894 (accessed on 9 April 2025).

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
