# Peer review of "Development of a BiAD Sensor for Locus-Specific Detection of Cellular Histone Acetylation Dynamics by Fluorescence Microscopy"

_genes, 2025, doi:10.3390/genes16040444_

Round 1

Reviewer 1 Report

Comments and Suggestions for Authors   The paper presents an interesting expansion of the BiAD sensor system with a specific use case of acetylation. We feel it is missing experiments that are key to establishing the system.
  1. A live-imaging experiment with the sensor in 293 cells with TSA treatment to demonstrate the feasibility of detecting changes over time, and not just TSA treatment then fixation;
  2. A second locus with a lower number of repeats that is closer to the detection limit (45 repeats) reported in the previous paper. It would be nice to see if the detection limit is universal across different BiAD modalities or if it is very dependent on what is being detected, as that would be another bottleneck to using this system.
  Minor points
  1. no sgRNA sequence reported, just the 15mer repeat;
  2. the figures are impossible to see because of their size, and red on black is hard contrast when the signal is dim;
  3. section 3.3/figure 3 does not actually image in living cells according to the methods unless there is a discrepancy there, or the section heading and description in the results/figure need to be clarified.
  4. Signal in -TSA in Figure vs Figure 3: Signal seems detectable in Figure 2 but not Figure 3. Why is this? Different detection settings? This should be made clear.

Author Response

Reply to reviewer 1

The paper presents an interesting expansion of the BiAD sensor system with a specific use case of acetylation. We feel it is missing experiments that are key to establishing the system.

Reply: Thank you for working with our manuscript and the helpful comments.

A live-imaging experiment with the sensor in 293 cells with TSA treatment to demonstrate the feasibility of detecting changes over time, and not just TSA treatment then fixation.

Reply: The BiAD technology is fully compatible with life cell imaging. This has been documented in our previous work and this important information has been added to the discussion. In our workflow, the sensors are active in living cells and we observed the time course of TSA treatment also in living cells. Only the readout of the sensor signals was done after cell fixation. This workflow allows to correlate epigenome signals directly with cell phenotypes or other cell stainings. The analysis of fixed cells provides several technical advantages in particular with respect to quantification and noise. Hence this workflow likely will be followed by many application cases of our technology. Within the scope of this work, it was not possible to conduct live cell imaging.

A second locus with a lower number of repeats that is closer to the detection limit (45 repeats) reported in the previous paper. It would be nice to see if the detection limit is universal across different BiAD modalities or if it is very dependent on what is being detected, as that would be another bottleneck to using this system.

Reply: As described in our paper, repetitive loci typically are not acetylated. Hence, we were lucky to find that the TTC34 locus for which a BiAD system was established shows acetylation. Unfortunately, currently we could not find another appropriate second candidate locus, hence the proposed experiments cannot be conducted. We agree that investigation of additional loci will be interesting and have mentioned this in the discussion of the revised manuscript.

Minor points

no sgRNA sequence reported, just the 15mer repeat

Reply: Thank you for catching this. The sgRNA sequence is now provided in the Methods section.

the figures are impossible to see because of their size, and red on black is hard contrast when the signal is dim;

Reply: The figures have been provided in high resolution. Perhaps the resolution was not sufficient in the proofs, which is not in our influence. In the final paper, readers will be able to zoom into them and see all details.

section 3.3/figure 3 does not actually image in living cells according to the methods unless there is a discrepancy there, or the section heading and description in the results/figure need to be clarified.

Reply: As mentioned above, the sensors are active in living cells and we observed the time course of TSA treatment also in living cells. Only the readout of the sensor signals was done after cell fixation. Therefore, the section heading referring to processes in living cells is correct.

Signal in -TSA in Figure vs Figure 3: Signal seems detectable in Figure 2 but not Figure 3. Why is this? Different detection settings? This should be made clear.

Reply: Please note that the comparison of the DMSO samples of the WT and mutant sensor in Figure 3D clearly documents a BiAD effect (p-value 6.0x10-3). This is also visible in the IFP2.0 signal in panel A after zooming into the figure. Hence, there is no disagreement between Figure 2 and 3. This has been described more clearly in the revised manuscript. Importantly in the context of the experiment reported in Figure 3, the BiAD signal and difference of the WT and mutant sensors becomes much larger after TSA treatment.

Reviewer 2 Report

Comments and Suggestions for Authors

Köhler et al. describe a biosensor system that detects the activating acetylation histone mark at a specific locus in live cells. This is based on their previous work in which they built a similar biosensor for monitoring repressive epigenetic marks. The authors make use of the BRD9 bromodomain as the reader/detector of histone acetylation at the TTC34 locus that is rich in repeats for the recruitment of the dCas9 anchor. With this work, the authors expand their repertoire of monitoring chromatin states, providing an important tool for probing specific genomic sites in live cells. The following points should be addressed, especially the first point to support the functionality of their histone acetylation biosensor.

  • The use of the binding-deficient BRD9 mutant as a negative control is commendable. However, it would be critical to use an orthogonal approach (e.g., H3K27ac ChIP-qPCR at the TTC34 locus) to show that the histone acetylation biosensor is specific. This would be especially valuable for the TSA experiment, as it would show how much TSA elevated H3K27ac levels at the TTC34 locus, and how ChIP-qPCR and biosensor approach correlate.
  • One of the biggest drawbacks of the BiAD sensor is the fact that it is currently restricted to repeats and therefore of limited usefulness to the community. Can the authors discuss how they envision their biosensor in the future could work at any given gene?
  • How long is the exposure time to detect the BiAD signal? Would the biosensor in its current iteration work in live cell imaging to record videos? This would be a valuable addition to the discussion.
  • What is the number of cells analyzed in each experiment? In addition, as the micrographs are quite small and possibly not of the best resolution in this version, could the outline of the nucleus be marked to help the reader orient themselves?

Author Response

Reply to Reviewer 2

Köhler et al. describe a biosensor system that detects the activating acetylation histone mark at a specific locus in live cells. This is based on their previous work in which they built a similar biosensor for monitoring repressive epigenetic marks. The authors make use of the BRD9 bromodomain as the reader/detector of histone acetylation at the TTC34 locus that is rich in repeats for the recruitment of the dCas9 anchor. With this work, the authors expand their repertoire of monitoring chromatin states, providing an important tool for probing specific genomic sites in live cells. The following points should be addressed, especially the first point to support the functionality of their histone acetylation biosensor.

Reply: Thank you for working with our manuscript and the helpful comments.

The use of the binding-deficient BRD9 mutant as a negative control is commendable. However, it would be critical to use an orthogonal approach (e.g., H3K27ac ChIP-qPCR at the TTC34 locus) to show that the histone acetylation biosensor is specific. This would be especially valuable for the TSA experiment, as it would show how much TSA elevated H3K27ac levels at the TTC34 locus, and how ChIP-qPCR and biosensor approach correlate.

Reply: Thank you for this comment. We have now validated the increase of histone acetylation after TSA treatment at the genome wide level by Western blot analysis. This is a more efficient procedure than ChIP-qPCR as it does not depend on the ChIP step with all its peculiarities.

One of the biggest drawbacks of the BiAD sensor is the fact that it is currently restricted to repeats and therefore of limited usefulness to the community. Can the authors discuss how they envision their biosensor in the future could work at any given gene?

Reply: We agree. This aspect is covered by the last paragraph of our discussion which has been further expanded in the revision.

How long is the exposure time to detect the BiAD signal? Would the biosensor in its current iteration work in live cell imaging to record videos? This would be a valuable addition to the discussion.

Reply: Thank you for this hint. The BiAD technology is fully compatible with recoding of videos. This important information has been added to the discussion.

What is the number of cells analyzed in each experiment? In addition, as the micrographs are quite small and possibly not of the best resolution in this version, could the outline of the nucleus be marked to help the reader orient themselves?

Reply: The number of analyzed cells has been added to the figure legends. The nuclei have been outlined as requested. The images have been provided at high resolution which will allow readers to zoom in at will.

Reviewer 3 Report

Comments and Suggestions for Authors

Köhler developed a bimolecular anchor detector (BiAD) sensor for locus-specific detection of histone acetylation based on the previous design of BiAD sensors. They integrated 2× BRD9 bromodomain (2xBRD9-BD) that binds to histone acetylation into a previously reported BiAD sensor and successfully developed a new type of BiAD sensor for histone acetylation. It is a concise short article and may shed light on future developments of new BiAD sensors for histone acetylation. The manuscript can benefit from resolving the following issues. 

Comments:

1. Multiple typos in the manuscripts, please fix them. TO name a few:

           (a) Lines 13-14: "It was integrated it into a dual-color BiAD chassis..."

           (b) Line 38: “...sensors in living cells cannot be connected with individual genes which limits the...”

2. Figure 2A: The strengths of the two signals seem to be different. But, if it is possible, please show a merged image to better show the colocalization of the signals.

3. Lines 130-133 of the methods section describe the quantification and normalization of BiAD signals. The relative BiAD signals in Figure 2C and 3D box plots are the normalized BiAD signals, right? Please indicate it in the y-axis titles.

4. Based on the design, only histone acetylation close to the target locus (YPet signal) can generate the IFP2.0 signal. However, in the fluorescent images, there are stronger IFP2.0 signals that are not close to the YPet signals. Where do those signals come from? Are they from non-specific binding of Histone acetylation? In addition, the Y57A mutants also show stronger IFP2.0 signals in the images.

5. What is the distribution of the BiAD signals? Are the images in Figures 2 and 3 from single cells? If possible, please indicate the cell margins.

Author Response

Reply to Reviewer 3

Köhler developed a bimolecular anchor detector (BiAD) sensor for locus-specific detection of histone acetylation based on the previous design of BiAD sensors. They integrated 2× BRD9 bromodomain (2xBRD9-BD) that binds to histone acetylation into a previously reported BiAD sensor and successfully developed a new type of BiAD sensor for histone acetylation. It is a concise short article and may shed light on future developments of new BiAD sensors for histone acetylation. The manuscript can benefit from resolving the following issues. 

Comments:

  1. Multiple typos in the manuscripts, please fix them. TO name a few:

(a) Lines 13-14: "It was integrated it into a dual-color BiAD chassis..."

(b) Line 38: “...sensors in living cells cannot be connected with individual genes which limits the...”

Reply: Thanks for noting this. These mistakes have been corrected.

  1. Figure 2A: The strengths of the two signals seem to be different. But, if it is possible, please show a merged image to better show the colocalization of the signals.

Reply: Please note that the key point is the presence of yellow spots with WT and mutant sensor (indicating correct sensor targeting), but absence of red spots in the mutant sensor (indicating absence of BiAD signal). This information is clearly visible in the current presentation. Merging the red and yellow channels does not improve visibility.

  1. Lines 130-133 of the methods section describe the quantification and normalization of BiAD signals. The relative BiAD signals in Figure 2C and 3D box plots are the normalized BiAD signals, right? Please indicate it in the y-axis titles.

Reply: The y-axis label indicates that relative signals are shown. We have now added to the figure legends, that they were normalized as described in the Method section.

  1. Based on the design, only histone acetylation close to the target locus (YPet signal) can generate the IFP2.0 signal. However, in the fluorescent images, there are stronger IFP2.0 signals that are not close to the YPet signals. Where do those signals come from? Are they from non-specific binding of Histone acetylation? In addition, the Y57A mutants also show stronger IFP2.0 signals in the images.

Reply: It is a well-documented problem in the split protein sensors that protein aggregation can occur leading to aberrant BiFC signals. This problem is overcome by using the dual color BiAD system, in which the yellow spots indicates the target locus allowing to restrict the analysis on these regions. This approach was introduced and validated in our previous paper, and it has been described in our introduction “The resulting dual-color BiAD sensors allow to detect the BiAD signal specifically at the locus of interest, which led to a strong improvement of signal-to-noise ratio, because background fluorescence fluctuation and non-specific protein aggregation could be excluded from the analysis.”

  1. What is the distribution of the BiAD signals? Are the images in Figures 2 and 3 from single cells? If possible, please indicate the cell margins.

Reply: Thank you for this suggestion. Cell nuclei have been indicated in all cells.

Reviewer 4 Report

Comments and Suggestions for Authors

The manuscript titled “Development of a BiAD Sensor for Locus-Specific Detection of Cellular Histone Acetylation Dynamics by Fluorescence Microscopy” by Köhler A.R.; et al. is a scientific work where the authors developed the next-generation of bimolecular anchor detector sensors from their previously reported works to sense single locus modifications in histone acetylation processes. Furthermore, a double acetylysine reader was tested to enhance the selectivity and specifity of the recognition of histone domains. The most relevant outcomes found by the authors could open promising gates in the design of future ultrasensitive detection devices for epigenomic studies. The manuscript is generally well-written and this is a topic of growing interest.

However, it exists some points that need to be addressed (please, see them below detailed point-by-point) to improve the scientific quality of the submitted manuscript paper.

1) The author should consider to add the term “bimolecular anchor detection” in the keyword list.

2) Introduction. “During the last years, we developed bimolecular anchor detector (BiAD) sensors to overcome this limitation (…) demonstrated that double reader domains generally perform better in BiAD applications (…)” (lines 43-64). Here, the authors highlight the relevance of the previously reported methodology to specifically detect the gene regulation processes by fluorescent bimolecular anchor detector sensors. It should be also neccesary to mention other bicolor fluorescent sensors developed in other fields [1,2]. This will strengthen the strategy pursued by the authors in this research.

[1] https://doi.org/10.1016/j.snb.2018.12.134

[2] https://doi.org/10.1039/d4lf00227j

3) Materials & Methods. This section is clearly explained. No actions are requested from the authors.

4) Results. “3.1. Identification of a Target Region for Histone Acetylation Detection with the Dual-Color BiaD Sensor” (lines 152-180). Did the authors found any challenge in the detection of the histone acetylation target regions based on their dynamic nature and continuous changing in response to cellular signals? Some insights need to be furnished in this regard.

5) “3.2. Integration of a Functional Acetyllysine Reader Domain into the Dual-Color BiAD Chassis” (lines 181-220). This comment is relatively linked to the previous point. Did the authors experience any crosstalk event among the acetyllysine reader domains with other histone regions? In case affirmative, what was the strategy pursued by the authors to minimize this effect?

6) “4. Discussion” (lines 248-282). This section perfectly remarks the most relevant outcomes found by the authors in this work and also the promising future prospectives in this field. It would be desirable to add a brief statement to state the potential future action lines to pursue the topic covered in this research.

Author Response

Reply to Reviewer 4

The manuscript titled “Development of a BiAD Sensor for Locus-Specific Detection of Cellular Histone Acetylation Dynamics by Fluorescence Microscopy” by Köhler A.R.; et al. is a scientific work where the authors developed the next-generation of bimolecular anchor detector sensors from their previously reported works to sense single locus modifications in histone acetylation processes. Furthermore, a double acetylysine reader was tested to enhance the selectivity and specifity of the recognition of histone domains. The most relevant outcomes found by the authors could open promising gates in the design of future ultrasensitive detection devices for epigenomic studies. The manuscript is generally well-written and this is a topic of growing interest.

However, it exists some points that need to be addressed (please, see them below detailed point-by-point) to improve the scientific quality of the submitted manuscript paper.

The author should consider to add the term “bimolecular anchor detection” in the keyword list.

Reply: Thank you for this good idea, this has been done.

Introduction. “During the last years, we developed bimolecular anchor detector (BiAD) sensors to overcome this limitation (…) demonstrated that double reader domains generally perform better in BiAD applications (…)” (lines 43-64). Here, the authors highlight the relevance of the previously reported methodology to specifically detect the gene regulation processes by fluorescent bimolecular anchor detector sensors. It should be also neccesary to mention other bicolor fluorescent sensors developed in other fields [1,2]. This will strengthen the strategy pursued by the authors in this research.

[1] https://doi.org/10.1016/j.snb.2018.12.134

[2] https://doi.org/10.1039/d4lf00227j

Reply: We have inspected these suggestions but found them to be unrelated to our work. Therefore, they were not cited. However, we have included the review „Studying Chromatin Epigenetics with Fluorescence Microscopy“ in our citation list now, which provides a detailed and well balanced background of the field. In addition, we have added a review on ChIP-seq (“ChIP-seq: advantages and challenges of a maturing technology”), to provide some additional background regarding the most widely used technology for the analysis of histone acetylation.

Materials & Methods. This section is clearly explained. No actions are requested from the authors.

Reply: Thank you for this favorable comment.

Results. “3.1. Identification of a Target Region for Histone Acetylation Detection with the Dual-Color BiaD Sensor” (lines 152-180). Did the authors found any challenge in the detection of the histone acetylation target regions based on their dynamic nature and continuous changing in response to cellular signals? Some insights need to be furnished in this regard.

Reply: HEK293 cells were cultivated under stable growth conditions and histone acetylation analyzed by ChIP-seq using standard approaches. With this approach histone acetylation was detected in the TTC34 regions. We did not analyze, if the acetylation levels at this locus show fluctuations under specific cultivation conditions.

“3.2. Integration of a Functional Acetyllysine Reader Domain into the Dual-Color BiAD Chassis” (lines 181-220). This comment is relatively linked to the previous point. Did the authors experience any crosstalk event among the acetyllysine reader domains with other histone regions? In case affirmative, what was the strategy pursued by the authors to minimize this effect?

Reply: This is an interesting question, but this cannot be studied with our current BiAD sensors, as they can only detect one modification a time. We have added this idea as a potential direction for future work at the end of the discussion section.

“4. Discussion” (lines 248-282). This section perfectly remarks the most relevant outcomes found by the authors in this work and also the promising future prospectives in this field. It would be desirable to add a brief statement to state the potential future action lines to pursue the topic covered in this research.

Reply: This has been covered at the end of the second but last paragraph of our discussion: „This problem could be overcome in future work by the generation of stable cell lines expressing the BiAD components encoded by plasmids 4 and 5 or by strategically combining multiple BiAD components on the same plasmid to make successful transfection of all plasmids visible.“

Round 2

Reviewer 1 Report

Comments and Suggestions for Authors

It does not seem any experimental critiques will be addressed in a meaningful manner. I suppose that is enough there for showcasing acetyl-BiAD

Author Response

Reply: This reviewer does not raise any specific points any more.

Reviewer 2 Report

Comments and Suggestions for Authors

My main concern was the specificity of the histone acetylation biosensor, which I feel was not adequately addressed. With their newly added H4 acetylation western blot, the authors show that their TSA treatment is working globally. However, they do not show how acetylation levels change at the TTC34 locus, which could be done by ChIP-qPCR. Based on their western blot result, acetylation levels likely increase more than 1,000-fold for both 3 h and 6 h TSA treatments. But their biosensor detected no change (3 h TSA) or only a 2-3-fold increase in acetylation (6 h TSA). If this disparity held true for the TTC34 locus, it would mean that their biosensor is not working, at least not quantitively. This important point could only be assessed by an orthogonal method such as ChIP-qPCR that shows the H4K12ac change specifically at the TTC34 locus upon TSA treatment.

Author Response

Reply: Please note that our conclusion that the newly developed BiAD Kac sensor is functional is based on well-documented and very solid data:

  • Acetylation at TTC34 locus (our test locus) has been experimentally shown in the cell line used in our experiments
  • Acetyl-specific histone binding of BRP9 and its loss by the Y57A mutation is well described in many papers cited in our manuscript
  • We observe specific BiAD signals with the WT sensor, but not with the completely identical BiAD sensor that just differs by a single Y57A mutation in both domains of the BRD9 double reader
  • We observed expected increases after TSA treatment in global histone acetylation levels
  • Our BiAD sensor (but not the mutant control sensor) detects an increase in Kac at the TTC34 locus

This line of argumentation has now been included in the discussion of the revision 2.

The reviewer now requests additional validation that histone acetylation is present at the TTC34 locus and it changes with TSA treatment. As mentioned above, ChIP-seq has already been done for this cell line, so the presence of histone acetylation at the TTC34 locus is experimentally documented. Repeating a full ChIP-seq experiment after TSA treatment just for the purpose of a validation that TSA treatment also works at the TTC34 locus is well beyond scope of this work and not justified.

The reviewer also proposed to use ChIP-qPCR, which we appreciated. However, despite considerable efforts we were not able to set up a reliable qPCR assay for the TTC34 locus most likely due it is repetitive sequence. Hence, ChIP-qPCR experiments could not be conducted for the tTC34 locus.

Finally, we like to mention that the estimation of the reviewer that the histone acetylation signal changed 1000-fold upon TSA treatment is incorrect. To document this better, we now also conducted a semiquantitative comparison of histone acetylation levels before and after TSA treatment (new Supplementary Figure 3C), showing that the global acetylation increases by about 10-fold. While this is still a larger change than observed at the TTC34 locus by the BiAD sensors, it needs to be considered that TTC34 is not a classical promoter or enhancer, where TSA treatment will trigger the strongest increase in histone acetylation. We conclude that the data are very consistent.